# Knowledge, attitudes, and practices (KAP) of nutrition among school teachers in Bangladesh: A cross-sectional study

**Mohammad Asadul Habib**[1☯*], **Mohammad Rahanur Alam**[1‡], **Tanjina Rahman**[2‡], **Akibul Islam Chowdhury**[1☯¤*], **Lincon Chandra Shill**[1]

**1** Department of Food Technology and Nutrition Science, Noakhali Science and Technology University, Noakhali, Bangladesh, **2** Institute of Nutrition and Food Science, University of Dhaka, Dhaka, Bangladesh

☯ These authors contributed equally to this work.
¤ Current address: Department of Nutrition and Food Engineering, Daffodil International University, Savar, Bangladesh
‡ MRA and TR also contributed equally to this work.
* akibul433@gmail.com (AIC); asadulhabib698@gmail.com, asadulhabib.ftns@nstu.edu.bd (MAH)

**Data Availability Statement:** All relevant data are within the manuscript.

**Funding:** The author(s) received no specific funding for this work.

## Abstract

### Background

Teachers play a pivotal role in imparting nutritional knowledge to their students. This research aimed to evaluate the nutrition-related knowledge, attitudes, and practices (KAP) of selected Bangladeshi school teachers across the country.

### Methods

A cross-sectional study was performed using a multistage sampling method. A pretested and structured questionnaire was used to collect data. Statistical analyses, including descriptive statistics, multiple regression analysis, and ANOVA tests, were performed to carry out the study.

### Results

Among the 280 participants, only 9.9% were happy with their understanding of childhood nutrition requirements, around 54.2% were familiar with basic nutrition-related knowledge, and overall, 97.7% of participants had a positive attitude towards learning about nutrition-related knowledge focusing on the well-being of children. Moreover, only 38.7% had training in pediatric nutrition. Age, type of school, type of residence, professional training of school teachers, and the intent of having ever taken part or paying attention to nutrition-related knowledge significantly impacted the respondents' nutrition knowledge score ($p < 0.05$).

### Conclusion

Adequate planning and intervention measures should be developed to improve teachers' understanding, behavior, and practice that encourage the growth of optimal nutrition-related behavior among school-aged children to establish a healthy nation.

**Competing interests:** The authors have declared that no competing interests exist.

## Introduction

*"Children are like buds in a garden and should be carefully and lovingly nurtured, as they are the nation's future and the citizens of tomorrow"*, is a famous quote by Jawaharlal Nehru, the first prime minister of India. Recent decades have seen a dramatic shift in social infrastructure and the rise of industrialized society, which have contributed to an alarming increase in childhood obesity rates across the developing world [1, 2]. Obesity is perhaps the most prevalent form of malnutrition in affluent countries, affecting adults and children. Children with poor eating habits, physical inactivity, and a poor social and economic environment are more likely to be overweight, malnourished, and develop chronic disease, cognitive disability, and non-communicable diseases (NCDs) in adulthood, such as diabetes, obesity, hypertension, metabolic syndrome, coronary heart disease, and so on [3, 4]. According to the recent data from the World Health Organization (WHO), childhood obesity has become one of this century's critical public health concerns [5]. It is reported that 20 to 40 percent of children and adolescents in wealthy countries are affected [6]. As of 2010, over 42 million children (<5 years) were facing overweight and nearly 35 million of those were from developing countries [7]. More than 22 million children under five are considered obese worldwide, with one out of every ten children being overweight. The prevalence of obesity in African and Asian countries is less (10%) than the European & American countries (20%) [8].

Both education and health are inextricably linked to one another [9]. The developing mind is more receptive to new information and experiences during childhood, making this a prime time for education that can increase the likelihood that knowledge and abilities will be retained and used later in life [10]. It is widely assumed that a child's capacity to reach their full potential is strongly tied to the beneficial effects of excellent health, proper nutrition, physical activity, and quality education [9]. Schools play a critical role in the community regarding health education and awareness. School does have the most important impact on the lives of children [11, 12]. Studies have also shown that school teachers can implement a successful student health-related education program [9]. Student behavior regarding diet can be changed by the teachers' ability to deliver nutrition education properly [13]. Trained teachers can quickly develop healthy eating habits among students, such as consuming fruits and vegetables, avoiding sugar-related products and unhealthy street foods, etc. [13, 14]. Some schools run education and health-related programs simultaneously to produce a "health-promoting" atmosphere that encourages learners to study [15]. According to several studies, healthy eating information in secondary school curricula is sparse, and even if it exists, it plays a minor role in the curriculum [5]. A nutrition-focused health education program at the school level is essential to address nutrition and health issues as the number of school-aged children, and health-related complications have risen in developing countries [16].

Various factors influence the development of excellent nutrition habits, and the knowledge–attitude–practice (KAP) paradigm provides a framework for bringing a constructive change in nutrition practices [17]. Improving the attitudes, knowledge, and practices regarding nutrition in children and adolescent is critical because it will result in a more food-conscious and healthier society. Teachers play an essential role in students' healthy dietary habits at an early life that may track down in adulthood via educating about more nutritious food choices, regular physical activity, and a sedentary lifestyle. They may also incorporate basic food and nutrition courses in the syllabus [13, 18]. In Bangladesh, there were no data about the teacher's knowledge regarding nutrition, however, the knowledge, attitude and practice of child, adolescent and mothers are evaluated [19–21]. So, there is a need to evaluate the knowledge, attitude and practice towards building an intense nutrition related perceptions among school teachers.

The current study aimed to evaluate the knowledge, attitudes, and practices (KAP) regarding nutrition among school teachers in Bangladesh. Determining their level of KAP for developing well-balanced nutritious eating habits and regular physical activity towards students will provide concrete evidence from which it could be possible to guide an intervention plan to improve the current condition.

## Materials and methods

### Study design

A cross-sectional study was conducted from February 2022 to June 2022 in different regions of Bangladesh. Multistage stratified random sampling was applied in collecting data from school teachers of nine (9) districts in Bangladesh (Fig 1). Initially, all respondents were given a self-administered standardized questionnaire [22]. Informed consent was taken from each participant and responses were only collected from school teachers who filled the questionnaire completely. A total of 280 school teachers were finally involved in this study. School teachers who did not provide their consent and did not fill the questionnaire were excluded. Additionally, no monetary reward or tangible prize was offered for their participation into the study.

### Data collection tools

This study used a pre-designed questionnaire based on the KAP model [22]. An android-based "KoBo Collect" software was used to collect the data. There were four sections of the

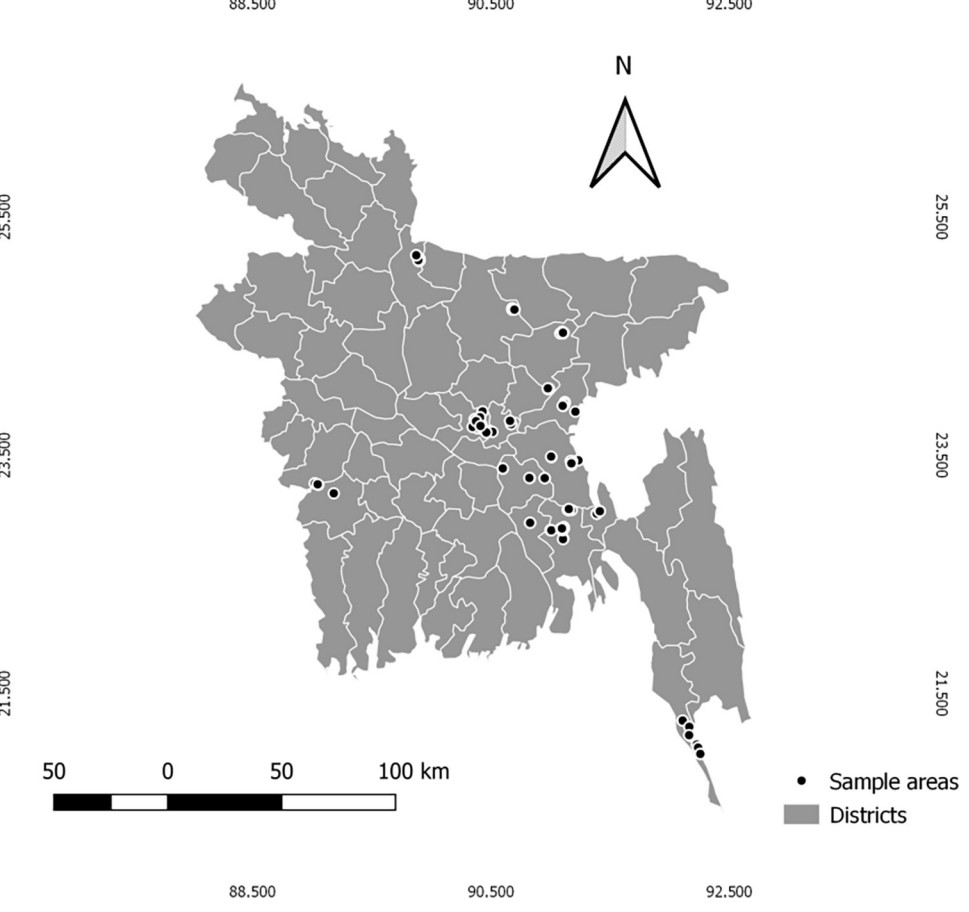

**Fig 1. Sample areas of the study (Prepared by using QGIS software version 3.10.2).**

questionnaire: *(a) demographic characteristics, (b) nutrition-related knowledge, (c) nutrition-related attitude,* and *(d) nutrition-related practices or behaviors*. A total of 66 questions were asked, of which **8** questions were *demographic*-related, **49** were *nutrition knowledge* related, **5** were *attitude* related, and **4** were *practice* related. The knowledge-related questions were divided into seven sections containing questions related to (i) protein, (ii) fat, (iii) vitamin, (iv) calcium, (v) dietary fiber, (vi) nutrient elements, and (vii) the child's nutrition.

## Scoring

The assignment technique was used to determine the nutrition-related knowledge score by explicitly judging each question as true or false. Answers were graded with a 1 for the correctness and a 0 for incorrectness. This section had a maximum score of 49 points. Each attitude question was categorized into three parts; (i) satisfied/confident/ willing, (ii) neutral, and (iii) unsatisfied/no confidence/ unwilling.

## Internal validity

The internal validity of the final questionnaire was measured by internal reliability, and Cronbach's alpha (α) was 0.742. The validity of three segments that evaluated the nutrition-related knowledge, attitude, and practice among school teachers was also measured by internal reliability, and Cronbach's alpha (α) was 0.773.

## Data analysis

The statistical analyses were performed using IBM SPSS 23.0 software.

The demographic features of survey respondents were used to compute descriptive statistics: frequency & percentage. To evaluate the mean differences among the continuous variables, ANOVA tests were used. A multiple linear regression analysis was performed to determine the influences on nutrition knowledge. Multiple linear regression analysis includes variables with statistical significance in single component analysis. A two-sided test was used to verify all statistics, and the statistical significance level was set at 0.05.

## Ethical approval

The study protocol was approved by the ethical board of Noakhali Science and Technology University, and respective institutes contributed to this study (NSTU/SCI/EC/2022/113). Informed written consent was also obtained from all participants before the study.

## Results

### Characteristics of school-teachers

Table 1 demonstrated the demographic characteristics of study participants, where we found that school teachers aged 17 to 57 years were mainly distributed at the age of >40 years (41.1%). Most of the teachers were male (72.9%); 81.8% of participants were married; 52.4% were from government schools, 9% were from urban areas, 35.4% have served as teachers for more than 10 years, and 45.4% have served as a school teacher for 3–10 years. About 57.1% had higher secondary education, and 42.9% had received above higher secondary education. Moreover, 78.9% of the respondents were professionally trained.

**Table 1. Demographic characteristics of participants.**

| Variables | N (%) | Variables | N (%) |
|---|---|---|---|
| **Age** | | **Education Level** | |
| **17–30 years** | 54 (19.3) | Primary | 0 |
| **31–40 years** | 111 (39.6) | Secondary | 0 |
| **>40 years** | 115 (41.1) | Higher | 160 (57.1) |
| | | Above | 120 (42.9) |
| **Gender** | | **Type of School** | |
| **Male** | 204 (72.9) | Government | 148 (52.9) |
| **Female** | 76 (27.1) | Private | 132 (47.1) |
| **Marital Status** | | **Working Time** | |
| **Unmarried** | 46 (16.4) | <3 years | 54 (19.2) |
| **Married** | 229 (81.8) | 3–10 years | 127 (45.4) |
| **Divorced** | 5 (1.8) | >10 years | 99 (35.4) |
| **Residence** | | **Professional Training** | |
| **Urban** | 162 (57.9) | Yes | 221 (78.9) |
| **Rural** | 118 (42.1) | No | 59 (21.1) |

Data were collected from a total of 280 school teachers distributed in nine districts throughout Bangladesh. Values were presented as numbers and percentages (N and %).

## Nutrition-related knowledge

Table 2 demonstrated the average scores of the school teachers on nutrition-related knowledge based on the demographic profile. The average score of nutrition-related knowledge was 34.33 (SD: 18.16), which was 70.07% of the total score of 49.00. The average scores (%) of the seven sections (Dimension **1** to Dimension **7***)* in Table 2 were as follows: (i) 63.23±15.15, (ii) 74.18± 17.22, (iii) 72.50 ± 13.22, (iv) 85.72±18.16, (v) 73.44± 15.55, (vi) 65.40±22.09, and (vii) 56.03 ±25.77. Interestingly, the *lowest average score* was found in "Child nutrition-related knowledge," and the *highest average score* was in "Calcium-related knowledge." In this analysis, gender (dimension 6), marital status (dimension 2), types of residence (dimension 2), an education level (dimension 1,2,3,4,5,6), type of school (dimension 5), working time (dimension 5,7), professional training (dimension 2,6) of school teachers, and total nutrition-related knowledge scores showed statistically significant differences ($p < 0.05$).

Table 2 also showed that the *highest score* was among participants *aged 31–40*, whereas the *lowest score* was found among teachers *aged 17–30*, *except for* "Protein-related knowledge." Participants who were married, residing in urban areas, or were more educated had a *higher average score* than other participants. Compared to teachers at private schools, government school teachers had higher average test scores. Participants in government schools received lower average scores than the total mean scores. Except for "Vitamin-related information," the average scores of the respondents who worked for more than ten years were the highest, while participants who worked for less than three years scored the lowest. School teachers having professional training perform better than those who don't have. However, untrained teachers' average scores were better than the scores for all students. "Protein-related knowledge" were statistically significant with only the education level of the demographic profile ($p<0.05$). "Fat-related knowledge" were statistically significant with demographic profile (marital status, residence, education level, professional training of teacher) ($p<0.05$). Vitamin-related knowledge and Calcium-related knowledge were statistically significant with only the education level of the demographic profile ($p < 0.05$). Dietary fiber-related information was statistically

**Table 2. Average score of each dimension of knowledge in different variables (mean, SD).**

| Variables | Dimensions | | | | | | |
|---|---|---|---|---|---|---|---|
| | 1 | 2 | 3 | 4 | 5 | 6 | 7 |
| Age (Years) | | | | | | | |
| *17–30* | 62.04±14.63 | 73.01±15.86 | 74.84±11.95 | 86.11±15.10 | 76.34±12.22 | 72.22±24.01 | 56.01±30.11 |
| *31–40* | 64.56±11.26 | 76.45±16.42 | 73.80±12.35 | 90.09±17.28 | 72.87±14.65 | 70.27±20.77 | 61.71±28.41 |
| *>40* | 64.93±14.20 | 76.02±15.89 | 73.77±13.30 | 87.61±19.13 | 76.52±15.70 | 68.98±21.04 | 55.87±30.0 |
| Gender | | | | | | | |
| *Male* | 64.22±13.21 | 75.35±16.86 | 74.51±13.02 | 87.99±18.81 | 75.05±14.91 | 71.73±20.43* | 58.95±29.77 |
| *Female* | 64.25±13.25 | 76.32±13.91 | 72.59±11.53 | 89.14±14.34 | 75.00±14.30 | 65.79±23.71* | 56.25±28.61 |
| Marital status | | | | | | | |
| *Unmarried* | 62.32±51.09 | 72.05±18.55 | 74.09±12.69 | 89.13±14.58 | 76.09±12.61 | 71.74±24.31 | 52.17±30.99 |
| *Married* | 64.62±12.89 | 76.29±15.59 | 74.05±12.74 | 88.43±18.36 | 75.16±14.85 | 70.16±20.88 | 59.60±29.15 |
| *Divorced* | 63.33±7.45 | 77.14±12.77 | 70.00±7.45 | 75.00±00.00 | 60.00±21.66 | 53.33±18.25 | 50.00±23.39 |
| Residence | | | | | | | |
| *Urban* | 64.71±12.78 | 73.28±15.28* | 73.25±12.41 | 88.27±16.05 | 73.79±15.39 | 69.75±21.60 | 56.40±30.11 |
| *Rural* | 63.56±13.77 | 78.81±16.68* | 75.00±12.94 | 88.35±19.79 | 76.74±13.63 | 70.62±21.43 | 60.69±28.41 |
| Education level | | | | | | | |
| *Primary* | 44.44±19.24 | 61.90±41.23 | 58.33±28.87 | 58.33±52.04 | 66.67±29.40 | 55.56±38.49 | 33.33±19.09 |
| *Secondary* | 65.35±14.41 | 61.90±23.18 | 61.11±16.31 | 78.57±26.56 | 59.26±19.98 | 53.97±26.82 | 54.17±23.83 |
| *Higher* | 65.10±12.65 | 74.20±16.06 | 74.01±11.11 | 87.66±16.82 | 73.96±13.07 | 70.00±20.88 | 61.09±27.11 |
| *Above* | 65.10±12.81 | 81.40±9.75 | 77.26±11.56 | 92.45±13.13 | 80.56±12.59 | 74.30±19.03 | 55.08±33.76 |
| Type of School | | | | | | | |
| *Government* | 64.18±12.35 | 76.83±14.09 | 75.06±12.62 | 88.51±16.85 | 76.66±14.47* | 17.72±18.20 | 56.25±31.40 |
| *Private* | 64.27±14.14 | 74.24±18.03 | 72.79±12.60 | 88.07±18.66 | 73.23±14.85* | 69.44±24.73 | 60.42±27.02 |
| Working time | | | | | | | |
| *<3 years* | 61.42±14.41 | 73.01±15.86 | 74.38±10.84 | 87.50±14.37 | 76.13±13.02* | 72.84±25.96 | 59.72±30.68* |
| *3–10 years* | 64.57±12.24 | 75.36±17.70 | 74.28±13.81 | 88.00±19.38 | 71.92±15.32* | 68.77±20.46 | 62.20±27.70* |
| *>10 years* | 65.32±13.61 | 77.34±13.84 | 73.40±12.0 | 89.14±17.19 | 78.45±14.09* | 70.37±20.14 | 52.27±30.21* |
| Professional training of teacher | | | | | | | |
| *Yes* | 64.47±12.67 | 77.18±14.70* | 74.67±11.58 | 89.37±17.02 | 75.87±14.44 | 71.64±19.34* | 58.89±29.47 |
| *No* | 63.27±15.09 | 69.73±19.50* | 71.46±15.87 | 84.32±19.64 | 71.94±15.49 | 64.40±27.58* | 55.72±29.39 |

significant with the education level, school type, and working time ($p < 0.05$). Nutrient element-related knowledge was statistically substantial with demographic profile (gender, education level, professional training of teacher) ($p < 0.05$) (Table 2). Child nutrition-related knowledge was statistically significant with only working time of the demographic profile.

## Nutrition knowledge learning-related attitudes

Approximately 61.8% of the school teachers reported having "*the confidence to conduct the childhood nutrition task properly*." Teachers who felt confident in themselves scored better than those who did not. Only 24.3% of interviewees reported being "satisfied" with their prior understanding of child nutrition. Teachers who were content with their present nutrition knowledge received lower marks than those who weren't. Regarding their current dietary expertise, the majority need to be more convinced. Approximately 81.8% of school teachers were "ready to get information linked to child nutrition knowledge using new media (Google, online journal, Facebook, what's app, Messenger and others)" in addition to 78.9% who were "willing to attend more childhood nutrition education and training." There were statistically significant variations with p-value <0.05 between attitude and behavior and nutrition-related knowledge (Table 3).

**Table 3. The average score of each dimension in nutrition-related attitudes.**

| Variables | N (%) | A score of attitudes (mean± SD) | P value |
|---|---|---|---|
| **A.1. Do you have the confidence to do the childhood nutrition work well?** | | | |
| *No confidence* | 26 (9.3%) | 49.62±18.86 | |
| *Neutral* | 81 (28.9%) | 67.78±14.23 | .000* |
| *Confidence* | 173 (61.8%) | 89.08±12.99 | |
| **A.2. Are you willing to learn more knowledge about children-related nutrition?** | | | |
| *Unwilling* | 10 (3.6%) | 39.00±20.79 | |
| *Neutral* | 45 (16.1%) | 58.89±17.99 | .000* |
| *Willing* | 225 (80.4%) | 85.11±13.99 | |
| **A.3. Are you satisfied with the childhood nutrition knowledge you already have?** | | | |
| *Satisfied* | 68 (24.3%) | 62.21±18.84 | |
| *Neutral* | 116 (41.3%) | 76.03±14.79 | .000* |
| *Unsatisfied* | 96 (34.3%) | 95.21±10.25 | |
| **A.4. Are you willing to attend more young childhood nutrition education and training?** | | | |
| *Unwilling* | 10 (3.6%) | 37.00±21.10 | |
| *Neutral* | 49 (17.5%) | 58.37±13.59 | .000* |
| *Willing* | 221 (78.9%) | 85.79±13.95 | |
| **A.5. Are you willing to get information related to children's nutrition knowledge through new media (Google, online journal, Facebook, what's app, Messenger and others)?** | | | |
| *Unwilling* | 11 (3.9%) | 40.91±24.68 | |
| *Neutral* | 40 (14.3%) | 58.25±17.95 | .000* |
| *Willing* | 229 (81.8%) | 84.76±14.04 | |

Data were collected from 280 school teachers distributed in nine different districts throughout Bangladesh. Values were presented as number and percentage (N and %) or mean and standard deviation (mean ± SD).

## Nutrition knowledge learning-related practices or behaviors

Only 42.5% of the participants took courses or received training in children's nutrition. Teachers who took classes or received training in childhood nutrition performed, on average better than those who did not. Concerning 67.9% of participants, nutrition awareness among youngsters was a focus. Teachers who were aware of the nutrition-related information of the students had better average results than those who weren't. Less than half (47.9%) of the participants have occasionally attempted to get their family members or acquaintances to learn more about the nutritional aspect of the children. The average scores of teachers were greater than those of those who frequently (12.1%), occasionally (47.9%), or never (32.9%) took the initiative to convey students' nutritional information to their families or friends. The vast majority of participants (53.4%) occasionally took the initiative to receive more knowledge on child nutrition. The likelihood that teachers will attempt to know about children's nutrition is higher than that of other adults: always (6.8%), frequently (15.4%), occasionally (52.6%), and never (24.3%). The following questions for the total nutrition-related knowledge scores revealed statistically significant differences with p value <0.05: "Have you participated in childhood nutrition knowledge courses or training?" "Have you paid attention to children's nutrition knowledge?" "Have you ever taken the initiative to promote children's nutrition knowledge to your relatives or friends?" "Have you ever taken the initiative to learn about child nutrition in your spare time?" (Table 4).

## Factors affecting nutrition-related knowledge

Age, school type, practice score, and nutrition-related knowledge among school instructors were all found significantly different in multiple linear regression models (p<0.05). With age,

**Table 4. The average score of each dimension in nutrition-related practices.**

| Variables | N (%) | A score of Practice (mean ± SD) | P Value |
|---|---|---|---|
| **P1. Have you ever participated in Childhood nutrition education knowledge courses or training?** | | | |
| *No* | 161 (57.5%) | 25.62±18.48 | .000 |
| *Yes* | 119 (42.5%) | 56.09±22.48 | |
| **P2. Have you ever paid attention to children's nutrition knowledge?** | | | |
| *No* | 90 (32.1%) | 16.53±16.10 | .000 |
| *Yes* | 190 (67.9%) | 49.01±21.85 | |
| **P3. Have you ever taken the initiative to promote children's nutrition knowledge to your relatives or friends?** | | | |
| *Never* | 92 (32.9%) | 14.13±13.89 | .000 |
| *Sometimes* | 134 (47.9%) | 40.86±11.38 | |
| *Often* | 34 (12.1%) | 63.97±10.55 | |
| *Always* | 20 (7.1%) | 92.50±13.69 | |
| **P4. Have you ever taken the initiative to learn about child nutrition in your spare time?** | | | |
| *Never* | 68 (24.3%) | 9.93±12.51 | .000 |
| *Sometimes* | 150 (53.6%) | 38.67±12.32 | |
| *Often* | 43 (15.4%) | 60.17±16.42 | |
| *Always* | 19 (6.8%) | 91.44±15.62 | |

Data were collected from 280 school teachers distributed in nine different districts throughout Bangladesh. Values were presented as number and percentage (N and %) or mean and standard deviation (***mean*** ± SD).

teachers became more knowledgeable regarding nutrition. Teachers from public schools had a greater understanding of nutrition than private schools. The teachers' working hours, professional backgrounds, and overall knowledge scores were statistically significant compared to their attitudes toward nutrition. Compared to teachers with less work time, those with more work time showed a more nutrition-related mindset. Those who had a more nutrition-related attitude scored higher in all areas of nutrition knowledge. School type, knowledge score overall, and nutrition-related practice among instructors were shown to differ significantly ($p<0.05$). Teachers who had attended public schools had more experience with nutrition. Those who had experience with nutrition-related activities scored higher overall in nutrition-related knowledge (Table 5).

**Table 5. Multiple linear regressions to identify factors that affect obesity-related knowledge, attitude & practice.**

| Demographic Variables | Knowledge | | | | Attitude | | | | Practice | | | |
|---|---|---|---|---|---|---|---|---|---|---|---|---|
| | B | SE | t | P | B | SE | t | P | β | SE | t | p |
| **Age (Years)** | -.723 | 1.140 | -.634 | .000* | 3.851 | 2.420 | 1.591 | .113 | -5.645 | 3.438 | .115 | .908 |
| **Gender** | .268 | 1.134 | .236 | .527 | -1.963 | 2.413 | -.814 | .417 | -2.617 | 3.430 | -1.642 | .102 |
| **Marital Status** | -.178 | 1.434 | -.124 | .813 | 2.151 | 3.053 | .705 | .482 | 5.915 | 4.328 | -.763 | .446 |
| **Residence** | .442 | 1.005 | .440 | .901 | 2.491 | 2.135 | 1.166 | .244 | -1.225 | 3.042 | 1.367 | .173 |
| **Education Level** | 5.165 | .820 | 6.296 | .660 | -.964 | 1.871 | -.515 | .607 | -5.619 | 2.639 | -.403 | .687 |
| **Type of School** | 1.591 | 1.065 | 1.494 | .000* | -.710 | 2.278 | -.312 | .755 | -2.928 | 3.233 | -2.129 | .034* |
| **Working Time** | .255 | 1.149 | .222 | .136 | -5.216 | 2.427 | -2.150 | .032* | .008 | 3.479 | -.905 | .366 |
| **Professional training of teacher** | -.519 | 1.378 | -.377 | .824 | -10.797 | 2.861 | -3.774 | .000* | .257 | 4.173 | .002 | .998 |
| **Total Knowledge Score** | Ref. | Ref. | Ref. | Ref. | .742 | .122 | 6.090 | .000* | .789 | .178 | 4.428 | .000* |
| **Attitude Score** | .163 | .027 | 6.090 | .707 | Ref. | Ref. | Ref. | Ref. | .105 | .086 | 1.213 | .226 |
| **Practice Score** | .086 | .019 | 4.428 | .000* | .052 | .043 | 1.213 | .226 | Ref. | Ref. | Ref. | Ref. |

Here, beta or β represents the standardized coefficient, a unit-free measure of effect size. 'SE' means the standard error of the regression. 't' represents the t statistics which is the coefficient divided by its standard error. A p-value of <0.05 is considered statistically significant.

## Discussion

The KAP model has been widely employed in public health concerning behavior-changing communication (BCC) strategies to encourage healthy behavior based on a society's local context. The guiding premise of this model was to increase knowledge since doing so will change attitudes and behaviors and lessen the socioeconomic toll that diseases have on society. According to this paradigm, an ideal diet could help with disease prevention, health promotion, and risk reduction [23]. The ability to adjust one's eating habits may be significantly influenced by knowledge [24].

This study revealed that the participating number of teachers with adequate understanding of nutrition were not satisfactory and overall, their correct score for all nutrition-related information was 70.07%. Without enough nutrition expertise, teachers won't be able to advocate health-related programs that may reduce childhood overweight and obesity related issues [25, 26]. Unlike educators in other nations, Bangladeshi teachers are found to bear better concerned about nutrition. Additionally, this study found that all domains had poor average scores, with "Child's nutrition-related knowledge" receiving the lowest score. The findings indicated that most of the participants with poor scores in this domain were deficient in nutrition related knowledge. Our study findings also revealed the existence of false beliefs and taboos on various nutrition related topics for children among school teachers, for instance, the state of child's health. The lack of understanding among teachers about child's nutrition seem to be a problem that may hinder their ability to raise awareness among their pupils. This may lead to poor health, which could hamper cognitive development and damage the learning environment, as those who have nutrition education and wellness opportunities can better teach their students about nutrition and aid in improving their health [27].

This survey is the first cross-sectional study to assess the association between nutrition-related knowledge and other aspects among school instructors, including attitudes and actions connected to nutrition knowledge learning and demographic variables. Here, we showed that factors including age, residence status, type of school, training of school teachers, the behavior of having ever taken part in training, and intent of ever paying attention to children's nutrition affect the overall nutrition-related knowledge.

The study further showed an increased awareness of school teachers with their age, which may be related to their extensive teaching experience and the ups and downs in various students' performances that they have seen. The study also found that private school teachers had the lowest scores in all domains of nutrition related knowledge. Previous studies on food-related views among rural American elementary and middle school teachers revealed that many lacked nutrition-related understanding regarding important foods [28]. Our investigations found that teachers with professional training (excluding nutrition courses) had less nutrition-related knowledge than the instructors who did not have a sort of training, probably because of their age and year of experience in service. According to our study, practices related to nutrition knowledge are associated with the total scores of nutrition related knowledge. Lack of adequate facilities and equipment for healthy food consumption in many schools may reduce the practice of healthy eating. Developing food and nutrition-based curricula in schools may improve the nutrition related KAP for students and teachers [29].

According to the results of the current study, 21.01% of the participants had never taken any training or course related to child's nutrition. Based on a recent survey, the percentage of child caregivers who attended workshops or played a role in service nutrition-related education was only one-third [30]. The current study also demonstrated that the teachers who took part in nutrition awareness training or courses had higher marks than those who did not. Some studies showed that teachers' training on nutrition and its related methods increases

their nutrition-related knowledge [31, 32]. However, better nutrition-related understanding cannot possibly be predicted by teachers' individualized attitudes and beliefs [33].

The current study also showed that all school instructors, regardless of their age with full coverage, must have their early training in strengthening nutrition-related knowledge. Most school teachers in our study, especially those with more nutrition-related expertise, exhibited good behaviors. Teachers' knowledge and food-related beliefs and behaviors can influence the habits of their student to adopt the healthy eating practices as teachers themselves could be role model for his students [34, 35].

School-based nutrition programs are required to encourage teachers to learn about nutrition actively. Our study also found that most respondents (81.8%) were willing to use social media (such as Facebook, WhatsApp, Google, online journals, and Messenger) to learn about child's nutrition knowledge. Additionally, people who were open to using these media had a higher likelihood of knowing something more related to nutrition than those who were neutral or opposed to it [36, 37]. However, further research is required on how preschool teachers could be informed through social media on issues related to public health, more particularly child health and nutrition. This way, it is possible to prevent child-hood obesity and malnutrition, the two most contemporary public health concerns, with the help of teachers.

This study had several limitations. First, we could not determine the causality due to the data from a cross-sectional survey. Second, the sample size of the study was small as the sub-jects were randomly selected based on 100% response rate that which may impact the precision and dependability of some of the findings. Furthermore, data on the knowledge, attitudes and nutrition practices among school instructors were self-reported. Finally, because there were only four and three items in the "nutrient element-related knowledge" and "calcium-related knowledge" sections of the questionnaire, perhaps, it is possible that the participants were nor familiar with the nutrition related topics that could bring information bias.

## Conclusion

A fair amount of nutrition expertise among school teachers was found only in a few districts (3 Out of 9 in this study) of Bangladesh. Most teachers were eager to learn about nutrition-related topics yet needed more formal training. In addition, nutrition-related knowledge among school teachers was correlated with age, sex, marital status, residence type, school type, educa-tional qualification, working hours, professional training, and attention to child health and nutrition information. Significant association was also found between the knowledge and behavior of the teachers about nutrition. The KAP model was used in this study to observe the existing condition and find out the gap in this field, which will help in planning effective inter-ventions to improve Bangladeshi school-teachers current understanding of child health and nutrition in future. School teachers can help in improving the nutritional status of children by adopting convenient, simple, cost-effective educational tools. They can also monitor both the physical (change in body weight for consuming a balanced diet) and psychological (change in attention span while studying) outcomes of their students. Thus, teachers could play a pivotal role in the overall well-being of our society.

## Acknowledgments

The author(s) would like to thank the post-graduated students from 5th Batch of the Food Technology and Nutrition Science department, Noakhali Science and Technology University for helping in data collection from different districts of Bangladesh.

## Author Contributions

**Conceptualization:** Mohammad Asadul Habib, Akibul Islam Chowdhury.

**Data curation:** Mohammad Asadul Habib, Akibul Islam Chowdhury.

**Formal analysis:** Mohammad Asadul Habib, Mohammad Rahanur Alam, Akibul Islam Chowdhury.

**Methodology:** Mohammad Asadul Habib, Mohammad Rahanur Alam, Akibul Islam Chowdhury, Lincon Chandra Shill.

**Supervision:** Mohammad Asadul Habib, Mohammad Rahanur Alam, Tanjina Rahman.

**Validation:** Mohammad Rahanur Alam, Tanjina Rahman.

**Visualization:** Akibul Islam Chowdhury, Lincon Chandra Shill.

**Writing – original draft:** Mohammad Asadul Habib, Tanjina Rahman, Akibul Islam Chowdhury.

**Writing – review & editing:** Mohammad Asadul Habib, Mohammad Rahanur Alam, Tanjina Rahman, Lincon Chandra Shill.

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
