## [Decision Letter · Decision Letter 0]

1 Feb 2023

PONE-D-22-31869Knowledge, Attitudes, and Practices (KAP) of Nutrition among School Teachers in Bangladesh: A Cross-Sectional StudyPLOS ONE

Dear Dr. Chowdhury,

Thank you for submitting your manuscript to PLOS ONE. After careful consideration, we feel that it has merit but does not fully meet PLOS ONE’s publication criteria as it currently stands. Therefore, we invite you to submit a revised version of the manuscript that addresses the points raised during the review process.

We look forward to receiving your revised manuscript.

Kind regards,

Benojir Ahammed, M.Sc.

Academic Editor

PLOS ONE

Journal Requirements:

Reviewers' comments:

Reviewer's Responses to Questions

**Comments to the Author**

1. Is the manuscript technically sound, and do the data support the conclusions?

Reviewer #1: No

Reviewer #2: Yes

2. Has the statistical analysis been performed appropriately and rigorously? 

Reviewer #1: Yes

Reviewer #2: Yes

3. Have the authors made all data underlying the findings in their manuscript fully available?

Reviewer #1: Yes

Reviewer #2: Yes

4. Is the manuscript presented in an intelligible fashion and written in standard English?

Reviewer #1: No

Reviewer #2: Yes

5. Review Comments to the Author

Reviewer #1: Thanks for the invitation and I am happy reading the paper.

I must say this is a well written manuscript the methodology is strong enough however I have some addressable comments.

First: The ethical approval of the paper is missing which is the strongest demerit of the paper for being rejected. Please add the approval number of your institution that had been given in this regard.

Second: the abstract is not organized and not even the style of PLoS one criteria.

Third: the rationality of the paper poor why this study important is missing in my sense.

Fourth: the generalazibility of the paper is poor due to its inherent study design and small sampling size.

Fifth: conclusion is overtly written please be specific based on your findings.

Sixth: how did you select covarites for final model that is missing.

Reviewer #2: Well written article. the study highlights the need for the teachers to be given more training. the analysis has been carried out nicely. the discussion and the results are aptly described. the conclusions are crisp.

6. PLOS authors have the option to publish the peer review history of their article (what does this mean?). If published, this will include your full peer review and any attached files.

Reviewer #1: No

Reviewer #2: No

---

## [Author Response · Author response to Decision Letter 0]

8 Feb 2023

Knowledge, Attitudes, and Practices (KAP) of Nutrition among School Teachers in Bangladesh: A Cross-Sectional Study

[PONE-D-22-31869]

I want to express my gratitude to the chief editor and the reviewers for their insightful comments and recommendations that I have already received. It was quite beneficial to consider new ideas in order to enhance the work. Please see below my detailed response to your comments and recommendations. I made an effort to take into account each advice that you so graciously made in light of my expertise and understanding.

Comments from Reviewer and Author’s responses:

Reviewer #1: 

Thanks for the invitation and I am happy reading the paper. I must say this is a well written manuscript the methodology is strong enough however I have some addressable comments.

1. The ethical approval of the paper is missing which is the strongest demerit of the paper for being rejected. Please add the approval number of your institution that had been given in this regard.

Response from Authors: Thanks to the reviewer for the valuable comment. We added the number of ethical approval (NSTU/SCI/EC/2022/113) given from the institution named Noakhali Science and Technology University, Bangladesh.

2. Second: the abstract is not organized and not even the style of PLoS one criteria.

Response from Authors: Abstract is organized according to the PLOS ONE style.

3. The rationality of the paper poor why this study important is missing in my sense.

Response from Authors: Thank you for your valuable comments and suggestions to this work. As per your suggestions the rationality in the background section is updated.

4. The generalizability of the paper is poor due to its inherent study design and small sampling size.

Response from Authors: Thank you for this comment. 

Based on response rate, we were able to keep only 280 school teachers in this study who filled up the questionnaire completely. Furthermore, in context of Bangladesh, the number of school teachers compare to the number of students is low and typically they have much work load. So, a lot of them are reluctant to participate in this study. Therefore, we recommend similar future study with large sample size to evaluate the whole scenario of the country.

5. Conclusion is overtly written please be specific based on your findings.

Response from Authors: We rewrite the conclusion focusing on our main findings along with the future recommendations as this study is the first study in Bangladesh evaluating teacher’s knowledge regarding nutrition.

6. How did you select covariates for final model that is missing.

Response from Authors: Thanks for this comment.

Selection of co-variates for final model is done by observing marginal association with the study’s outcome which is determined by a previous pilot study. We also considered the previous literature published in other countries to select the covariates for our study.

Reviewer#2:

Well written article. the study highlights the need for the teachers to be given more training. the analysis has been carried out nicely. the discussion and the results are aptly described. the conclusions are crisp.

Response from Authors: Thanks to you for your valuable observations, comments and recommendations.

---

## [Decision Letter · Decision Letter 1]

12 Mar 2023

Knowledge, Attitudes, and Practices (KAP) of Nutrition among School Teachers in Bangladesh: A Cross-Sectional Study

PONE-D-22-31869R1

Dear Dr. Chowdhury,

We’re pleased to inform you that your manuscript has been judged scientifically suitable for publication and will be formally accepted for publication once it meets all outstanding technical requirements.

Kind regards,

Benojir Ahammed, M.Sc.

Academic Editor

PLOS ONE

Additional Editor Comments (optional):

Reviewers' comments:

Reviewer's Responses to Questions

**Comments to the Author**

1. If the authors have adequately addressed your comments raised in a previous round of review and you feel that this manuscript is now acceptable for publication, you may indicate that here to bypass the “Comments to the Author” section, enter your conflict of interest statement in the “Confidential to Editor” section, and submit your "Accept" recommendation.

Reviewer #1: All comments have been addressed

Reviewer #2: All comments have been addressed

2. Is the manuscript technically sound, and do the data support the conclusions?

Reviewer #1: Partly

Reviewer #2: Yes

3. Has the statistical analysis been performed appropriately and rigorously? 

Reviewer #1: Yes

Reviewer #2: Yes

4. Have the authors made all data underlying the findings in their manuscript fully available?

Reviewer #1: Yes

Reviewer #2: Yes

5. Is the manuscript presented in an intelligible fashion and written in standard English?

Reviewer #1: Yes

Reviewer #2: Yes

6. Review Comments to the Author

Reviewer #1: Thanks to the authors for addressing my previously did comments. Now it is rideable and in better fashion however the conclusion can be rewritten since it is overtly written.

Reviewer #2: (No Response)

7. PLOS authors have the option to publish the peer review history of their article (what does this mean?). If published, this will include your full peer review and any attached files.

Reviewer #1: No

Reviewer #2: No

---

## [Editor Report · Acceptance letter]

15 Mar 2023

PONE-D-22-31869R1 

Knowledge, attitudes, and practices (KAP) of nutrition among school teachers in Bangladesh: a cross-sectional study 

Dear Dr. Chowdhury:

I'm pleased to inform you that your manuscript has been deemed suitable for publication in PLOS ONE. Congratulations! Your manuscript is now with our production department. 

Kind regards, 

on behalf of

Mr. Benojir Ahammed 

Academic Editor

PLOS ONE